# Efficacy of Adipose-Derived Mesenchymal Stem Cells and Stromal Vascular Fraction Alone and Combined to Biomaterials in Tendinopathy or Tendon Injury: Systematic Review of Current Concepts

**DOI:** 10.3390/medicina59020273

**Published:** 2023-01-31

**Authors:** Letizia Senesi, Francesco De Francesco, Andrea Marchesini, Pier Paolo Pangrazi, Maddalena Bertolini, Valentina Riccio, Michele Riccio

**Affiliations:** 1Department of Plastic Reconstructive Surgery, Hand Surgery Unit, Azienda Ospedaliero Universitaria delle Marche, 60121 Ancona, Italy; 2Orthopaedic and Traumatology 2 Surgery for the Upper Limb and Reconstructive Microsurgery, Azienda Ospedaliero Universitaria Città della Salute e della Scienza, 10126 Turin, Italy; 3School of Bioscience and Veterinary Medicine, University of Camerino, 62024 Matelica, Italy

**Keywords:** adipose-derived mesenchymal stem cells, stromal vascular fraction, biomaterials, tendon injury, tendinopathy

## Abstract

*Background and Objectives:* Tendon injury and tendinopathy are among the most frequent musculoskeletal diseases and represent a challenging issue for surgeons as well as a great socio-economic global burden. Despite the current treatments available, either surgical or conservative, the tendon healing process is often suboptimal and impaired. This is due to the inherent scarce ability of tendon tissue to repair and return itself to the original structure. Recently, Adipose-derived mesenchymal stem cells (ADSC) and stromal vascular fraction (SVF) have gained a central interest in the scientific community, demonstrating their effectiveness in treatments of acute and chronic tendon disorders in animals and humans. Either enzymatic or mechanical procedures to obtain ADSC and SVF have been described and used in current clinical practice. However, no unified protocols and processes have been established. *Materials and Methods:* This systematic review aims at providing a comprehensive update of the literature on the clinical application of ADSC enzymatically or mechanically processed to obtain SVF, alone and in association with biomaterials in the local treatment of tendinopathy and tendon injury in vivo, in animal models and humans. The study was performed according to the Preferred Reporting Items for Systematic Reviews and Meta-Analyses (PRISMA). *Results:* Thirty-two articles met our inclusion criteria, with a total of 18 studies in animals, 10 studies in humans and 4 studies concerning the application of biomaterials in vivo in animals. The review of the literature suggests that ADSC/SVF therapy can represent a promising alternative in tendonregenerative medicine for the enhancement of tendon healing. *Conclusions:* Nevertheless, further investigations and randomized control trials are needed to improve the knowledge, standardize the procedures and extend the consensus on their use for such applications.

## 1. Introduction

The healing process after tendon injury or tendinopathy still presents a challenging task for physicians. In fact, the natural tendon repair process is inefficient, elaborate and slow [1]. This is caused by the limited inherent healing ability of tendon tissue after trauma: the poor cellular population and vascularization cause an insufficient extracellular matrix (ECM) remodeling associated to a higher collagen-3 (COL-III) deposition in respect to collagen-1 (COL-I) (about two to three times more) [2]. Moreover, several inflammatory mediators are involved and intensified in tendinopathies, as well as in the engagement of growth factors [3]. All these processes taken together invalidate tendon repair [4,5].

The high worldwide incidence of these pathologies also represents a financial challenge. It is estimated that over 30-million human tendon/ligament-related procedures take place annually, representing an expenditure of over EUR 150 billion in American and European countries [6,7,8].

Unfortunately, current treatments available, either conservative [9,10,11,12,13,14] or surgical [15,16,17], are unable to restore the original tendon structure, functionality and biomechanical features [18,19]. Therefore, new treatments with the aim of improving tendon regeneration are necessary.

Mesenchymal stem cells (MSC) have gained, over time, an increasing interest in the scientific community. They have three characteristics: staying quiescent/undifferentiated until activated, being capable of differentiation into multiple tissue types, and continual cell replications/self -renewal [20].

Among the MSCs, Adipose-derived stem cells (ADSCs) represent probably the most attractive cell source [21] because they are multipotent and can be isolated from subcutaneous adipose tissue and from liposuction aspirates [22,23] in large quantities with minor donor site morbidity and discomfort.

Zuk et al. described in 2001 for the first time that Stromal Vascular Fraction (SVF) enzymatically obtained from adipose tissue contains large numbers of MSC cells that are able to differentiate into adipogenic, chondrogenic, myogenic and osteogenic lineages [24].

Nevertheless, Good Manufacturing Practice regulations affirm that the extensive use and manipulation of stem cells are not admissible for human use in European countries, according to the European Parliament and Council (EC regulation 1394/2007) related to minimal tissue manipulation [25,26].

Therefore, some authors demonstrated that alternative mechanical procedures, as compared to the original enzymatic method, are equally able to isolate SVF [26,27] and contain a heterogeneous mixture of cells, including endothelial cells, smooth muscle cells, pericytes, fibroblasts, mast cells, pre-adipocytes and a rich source of ADSCs [28].

Several studies demonstrated that SVF and ADSCs seem to improve tendon healing [1,6,29,30,31,32,33]. Therefore, the use of autologous ADSCs as therapy is feasible and has been shown to be safe and efficacious in preclinical and clinical studies.

The aim of this specific systematic review is to outline the current findings about ADSC enzymatically or mechanically processed to obtain SVF, alone and in association with biomaterials in the local treatment of tendinopathy and tendon injury in vivo in animal models and humans.

## 2. Materials and Methods

### 2.1. Search Strategy

(i) Search site: Articles are from PubMed, a database of papers on biomedical science. (ii) Database: MEDLINE(Pubmed), the Cochrane Library, EMBASE and Scopus. (iii) Tendon injury, tendinopathy (iv) Boolean algorithm: (“Adipose-derived mesenchymal stem cells” OR “stromal vascular fraction”) AND (“Tendinopathy” OR “Tendon injuries”) AND (“Biomaterials”). (v) Retrieval timeframe: We searched the selected journals published from 2012 to 2022. (vi) Inclusion and exclusion criteria: The search process was performed as PRISMA flow diagram (Figure 1).

### 2.2. Data Extraction and Analysis

Two observers (L.S. and F.D.F.) independently searched and collected data from the included studies. Any discordances were solved by consensus with a third author (M.R.). All data concerning adipose stem cell harvesting, manipulation and application were carefully reviewed and collected. Numbers software (Apple Inc., Cupertino, CA, USA) was used to tabulate the obtained data.

## 3. Results

According to the requirements of this specific systematic review, 32 papers have been selected and analyzed.

### 3.1. Animal Study

#### 3.1.1. Enzymatically Derived ADSC-SVF in Tendinopathy and Tendon Injury

The search study yielded 17 results and are listed in Table 1. Three studies were performed in iatrogenic injured Achilles tendons in rats. Lee and colleagues [34] obtained ADSC isolation from the lipoaspirate of human subcutaneous fat tissue of healthy donors, and after the enzymatic process, ADSCs were implanted in Sprague Dawley rats with full-thickness rectangular defects in the Achilles tendon. They performed histological, immunohistochemical and proteomic evaluations. Compared to the control group, ADSC therapy led to better gross morphological and biomechanical recoveries than those in both the fibrin and sham groups. Moreover, an increased expression of human-specific COL-I and tenascin-1 was observed. Oshita et al. [21] investigated the effects of ADSCs on Achilles tendon healing in 16 F344/NSIc rats that underwent collagenase injections in the Achilles tendon to simulate tendinopathy. They obtained ADSC from the rats themselves and performed histological and immunohistochemical evaluations. At an established follow-up, the authors determined that the ADSC group showed a significantly lower degree of tendon degeneration than the saline group. Furthermore, the Type III-to-Type I collagen ratio was significantly lower in the ADSC group, and this continued to decrease relative to the saline group. De Aro et al. [35] investigated the effects of ADSCs combined with growth derived factor-5 (GDF-5) in a transected Achilles tendon model in mice and performed genic and biomechanics evaluations. They determined an increase in the organization of collagen fibers in the injury-adjacent region in the ADSC group with an enhanced expression of Lisyl Oxidase (Lox), Decorin (Dcn) and Trasforming Growth Factor-β1 (TGF-β1) compared to other groups. Moreover, at biomechanical analysis, tendons treated with ADSC were more resistant to traction, with lower deformation at higher stress.

Two studies were performed in the iatrogenic rotator cuff in mice. Chen et al. [36] harvested ADSC from human healthy donors and injected it in injured supraspinatus tendons in rats; they performed histological, genic and biomechanical evaluations. An immediate enhancement in the healing process was found in the ADSC group at seven days, although no differences in the biomechanics or the histology between ADSC-treated groups and control groups were shown at other time points. Valencia Mora et al. [37] performed histological and biomechanical examinations in sovraspinatus rotator cuff tendons that were detached and sutured, treated with collagen and collagen+ADSC and compared to the control group. The histological examination showed less acute inflammation with a diminished presence of oedema and neutrophils in the ADSC group, while no differences in biomechanics results were found among other groups.

Two studies were performed in dogs: one in flexor tendon injuries [38] and one in rotator cuff injuries [39]. Shen et al. performed Zone II flexor tendon transections and sutures, wrapping the tendon with an ADSC sheet and collagen. They evaluated the inflammatory response using gene expression assays, immunostaining and histological assessments. ADSC was able to modulate the inflammatory phase of tendon healing, as demonstrated by gene expression and immunohistological assessments. In particular, ADSCs promoted a regenerative/anti-inflammatory M2 macrophage phenotype and influenced tendon extracellular matrix remodeling, angiogenesis and cell survival [38]. Canapp et al. [39] injected autologous ADSC and platelet-rich plasma (PRP) in dogs for the treatment of supraspinatus tendinopathy. They evaluated outcomes through Ultrasound (US), Magnetic Resonance Imaging (MRI) evaluation and final gait analysis. After 90 days, a significant increase in the total pressure index percentage (TPI%) was noted in the treated forelimb. Among the cases receiving ADSC therapy, 88% did not show significant differences in TPI% between the injured and the healthy forelimb, while the rest experienced significant improvement.

Six studies were performed in rabbit models, including three in Achilles tendon iatrogenic injuries. Vieira et al. were the first investigation team that treated ruptured Achilles tendons of rabbits with ADSC injection. They performed a histological analysis that showed a significant increase in the capillaries and in the structural organization of collagen in the ADSC-treated group compared with the surgically sutured group [40]. Chen et al. [41] investigated the tendon regenerative ability of ADSC in a rabbit Achilles tendon transection/repair model. In this study, the effects of ADSC on the proliferation and migration of tenocytes in vitro and on the biomechanical strength and protein expression of repaired tendons in vivo were investigated. Final findings indicated that ADSCs promote tenocyte proliferation and improve the mechanical properties of the repaired tendon. Therefore, they promoted tendon regeneration by upregulating tendon-specific markers in surgically repaired tendons. Uysal et al. investigated the biomechanical and immunohistochemical effects of the local administration of ADSCs in tendon repair. They showed an increase in tensile strength, a direct differentiation of ADSCs into tenocytes and increases in angiogenic GFs [42]. Lu et al. [43] performed bilateral amputation of the supraspinatus tendon in rabbit models and parallel reconstruction, injecting ADSC into the tendon-bone interface. They performed histological, immunohistochemical, biomechanical and MRI evaluation studies. ADSC injection into the shoulder cuff insertion tendon–bone interface can accelerate the formation of the “tidal line” at 8 weeks after operation, which is similar to the structure of the direct insertion point (bone–cartilage transitional zone), accelerate the transformation of collagen fiber types, and promote the Bone Morphogenetic Protein-2 (BMP-2) expression to a certain extent. There was also an increase of biomechanical results and MRI signaling, with better results after 12 weeks.

Two studies are about flexor tendon injuries in rabbits. Behfar et al. in 2011 [28] performed histological, immunohistochemical and biomechanical studies in the transection of deep digital flexor tendons. The results revealed a superior fibrillar continuity in treated groups over control groups. In addition, there was an increased expression of COL-I in SVF-treated tendons, while no changes in the COL-III (intensity and localization of the immunostaining) expression were found between the groups. Moreover, tensile strength parameters were significantly higher in SVF-treated tendons compared with the control groups. The same authors performed another biomechanical study in 2012 that showed significant increases in ultimate and yield load, energy absorption and stress in the treated group [44].

Three studies were performed in the equine model in superficial digitorum flexor tendon (SDFT) iatrogenic lesions. Geburek et al. evaluated through histological, immunohistochemical, biomechanical and sonographic analysis in groups treated with autologous inactivated serum and ADSC. The authors did not find statistical differences between the control and treated groups at the same time point for fiber arrangement, scores for metabolic activity and sub-scores for fiber structure and alignment. Moreover, no differences were found for biomechanical and sonographic analysis [45]. Carvalho et al. [46] performed histological, immunohistochemical and US evaluation in enzymatic-induced tendinopathy of SDFT. The histological evaluation demonstrated that ADSCs, combined with a platelet concentrate therapy, resulted in a better organization of collagen fibers and a decrease of the inflammatory infiltrate. In addition, the US evaluation showed a lack of lesion progression in the treated group. Conze et al. [47] performed histological, immunohistochemical and US doppler evaluations and showed that ADSC injection has a beneficial effect on the neovascularization of healing tendons. Histology staining was significantly higher for new vessels, and immunohistochemistry showed a higher expression of FVIII compared to the control groups. Lastly, Polly et al. [48] performed an in vitro study after the intralesional injection of SVF cells in a collagenase-induced tendon injury model in the horse. Both SVF and ADSC expressed growth factors important in tendon healing, such as Insulin growth factor-1 (IGF-1), TGF-β1 and Stromal cell derived factor-1 (SDF1), while a lower level of Cartilage Oligomeric Matrix Protein (COMP) is shown in the treated group.

**Table 1 medicina-59-00273-t001:** Enzymatically derived ADSC/SVF in animal model tendinopathy and tendon injury.

Authors	Animal Model	Injury	ADSC-SVF Provenience	Treatment	Investigation	Outcomesx
Lee et al. [34]	Sprague Dawley rats-in vivo	Full thickness Achilles tendon defect	Human subcutaneous fat tissue	Group 1: fibrin glue-ADSC injection Group 2: fibrin glue Group 3: control group	Histological Immunohistochemical, Biomechanical, Proteomic evaluation	Better gross morphological and biomechanical recovery in ADSC group. Increased expression of HST-C1 and T-1.
Oshita et al. [21]	16 F344/NSIc rats	Collagenase iatrogenic Achilles tendon injury	inguinal fat pads of two F344/NSlc rats	Group 1: ADSC injection Group 2: saline injection	Histological Immunohistochemical	ADSC group decreased levels of disrupted collagen fibers, cellularity, and hypervascularity. Increased expression COL1, decreased expression COL3.
De Aro et al. [35]	Lewis rat	Transected Achilles tendon	Inguinal region of 10 male Lewis rats	Group 1: untreated Group 2: ADSC Group 3: GDF-5 Group 4: ADSC + GDF-5	Biomechanical, RT-PCR	The ADSC increased expression of Lox, Dcn, and Tgfb1. Lower deformation at higher stress.
Chen et al. [36]	Sprague-dawley rats	Collagenase induced rotator cuff injury	Human subcutaneous fat tissue	Group 1: untreated Group 2: ADSC	Histological, RT-PCR, Biomechanical	Improvement in collagen fibres alignment, increased COL1, TNC expression No biomechanical differences at final endpoint
Valencia Mora et al. [37]	Sprague-Dawley rats	Detachment and repair of the supraspinatus tendon	Rat subcutaneous fat tissue	Group 1: collagen carrier Group 2: collagen carrier + ADSC Group 3: untreated	Biomechanical Histological	ADSC group showed less acute inflammation signs. No significative differences in biomechanical analysis.
Shen et. al. [38]	Mongrel Dogs	Second and fifth flexor digitorum profundus (FDP) zone II tendon transection and repair	Autologous subcutaneous fat tissue+ collagen sheet	Group 1: ADSC + collagen sheet wraps around the tendon Group 2: HA Group 3: suture only	Gene expression Immunostaining Histological	ADSC modulated the inflammatory phase of tendon healing, promoted a regenerative/anti-inflammatory M2 macrophage phenotype and influenced tendon extracellular matrix remodeling, angiogenesis, and cell survival.
Canapp et al. [39]	Dogs	Supraspinatus tendinopathy	Autologous subcutaneous fat tissue+PRP	Group 1: ADSC + PRP peritendinous injection by US guidance Group 2: contralateral non-affetcted	MRI, US, XR evaluation; Gait analysis	TPI significant increase in Treated limb
Vieira et al. [40]	New Zeland Rabbits	Achilles tendon cut injury	Autologous subcutaneous adipose tissue	Group 1: tendon cut Group 2: tendon suture Group 3: tendon suture + ADSC	Histological Analysis	Increase of vascular score and collagen fibres structural organization in Group 2 and 3
Chen et al. [41]	New Zeland Rabbits	Achilles tendon cut injury	Autologous subcutaneous adipose tissue	Group 1: tendon cut and sutured Group 2: tendon cut and sutured + ADSC	Cell viability, histological, proteomic analysis Biomechanical analysis	Increased tenocyte viability in ADSC group, mechanical strength and increased expression of TM BG Dcn
Uysal et al. [42]	Japanese rabbits	Achilles tendon cut injury	Autologous subcutaneous adipose tissue	Group 1: PRP gel Group 2: PRP + ADSC	Histological immunohistochemical, biomechanical analysis	COL1 FGF VEGF higher in ADSC treated group. Tgfb lower. Higher tendon tensile strength in ADSC treated group.
Lu et al. [43]	New zeland rabbits	Supraspinatus tendon incision and suture	Autologous subcutaneous adipose tissue	Group 1: ADSC-FG Group 2: FG	Histological, immunohistochemical, biomechanical analysis, MRI analysis	Higher expression of COL1, and ratio of COL3/COL1. Higher biomechanical value in ADSC-FG group, MRI better results after 12 weels
Behfar et al. [28]	New zeland rabbits	Deep digital flexor tendon transection and sutured.	Autologous subcutaneous adipose tissue	Group 1: tendon suture Group 2: tendon suture + ADSC	Histological, immunohistochemical, biomechanical analysis	Higher orientation of collagen bundles and fibrillar continuity. Increased expression of COL1. Higher tensile strength parameters.
Behfar et al. [44]	New zeland rabbits	Deep digital flexor tendon transection and sutured.	Autologous subcutaneous adipose tissue	Group 1: tendon suture Group 2: tendon suture + ADSC	biomechanical analysis	Significant increases in ultimate and yield load, energy absorption, and stress were noted at both time points when treatment groups were compared to their matched controls
Geburek et al. [45]	Warmbloods and trotter horses	Superficial digital flexor tendon surgical core lesion	Autologous subcutaneous adipose tissue	Group 1: inactivated autologous serum Group 2: ADSC injection	Histological, immunohistochemical biomechanical, clinical, US examination	No differences in histological findings, no difference in GAG, biomechanical, US results.
Carvalho et al. [46]	Mixed breed horses	Superficial digital flexor tendon collagenase iatrogenic injury	Autologous subcutaneous adipose tissue	Group 1: control Group 2: ADSC injection	Flow cytometric peripheral blood fluorescent microscopy	Cell presence, nanocrystals valuable in vivo markers
Conze et al. [47]	Warmblood and standardbreed horses	superficial digital flexor tendon surgical core lesion	Autologous subcutaneous adipose tissue	Group 1: control Group 2: ADSC injection	Histological, immunohistochemical, US analysis	Histology staining significantly higher for new vessels and FVIII. More vascularization in us doppler control.
Polly et al. [48]	Mixed breed horses	Superficial digital flexor tendonitis	Autologous subcutaneous adipose tissue	Group1: control Group2: ADSC co-cultured	Genic expression analysis	Significantly higher expression of IGF1, Tgfb1, SDF1. Lower expression of COMP.

Adipose-Derived Stem Cell (ADSC); Stromal Vascular Fraction (SVF); Human-specific Type I collagen (HST-C1); Tenascin-1(TNC); Collagen Type 1 (COL1); Collagen Type 3 (COL3); Growth differentiation factor-5 (GDF-5); Real Time-PCR (RT-PCR); Lysyl oxidase (Lox), Decorin (Dcn); Transforming growth factor, beta 1 (Tgfb1); Hyaluronic Acid (HA); Platelet Rich Plasma (PRP); Ultrasound (US); Magnetic Resonance Imaging (MRI); X-rays (XR); Cross Sectional Area (CSA); Total Pressure Index (TPI); Tenomodulin (TM); Biglycan (BG); Fibroblast Growth Factor (FGF); Vascular Endothelial Growth Factor (VEGF); Fibrin glue (FG); Glycosaminoglycan (GAG); Factor VIII(FVIII); Insulin like growth factor1 (IGF1); SDF1 Stromal derived factor1 (SDF1); Cartilage Oligomeric Matrix Protein (COMP).

#### 3.1.2. Mechanically Derived ADSC-SVF in Tendinopathy and Tendon Injury

Only one study in the literature reported mechanically isolated SVF to treat tendinopathy in SDFT in sheep [19]. The lesion was induced iatrogenically by collagenase injection. An in vivo and an in vitro evaluation were yielded. Histological immunihistochemical, RT-PRC, US, elastosonography and biomechanical tests were performed. ADSC administration was able to maintain a higher organization of fibres, angiogenesis and collagen deposition. Elastosonography evaluation showed a higher hardness of a treated tendon compared to a control one. The expression of regenerative factors COL-I, FVIII, vimentin were higher in the treated group compared to the control group. In contrast, COL-III and TGF-β1 (considered scarring factors) were higher in the control group. No differences in the biomechanical analysis were shown.

Taken together, all the examined studies performed in animals showed a significant improvement of the morphological tendon aspect, which was revealed by histological analysis. Moreover, all the studies are coherent about proteomic and genetic analysis, showing a “normal tendon-like” collagen expression after ADSC administration. Moreover, another common issue is the anti-inflammatory features after ADSC administration.

On the contrary, biomechanical, US and MRI analysis did not reveal uniform improvement.

### 3.2. Human Study

#### 3.2.1. Enzymatically Derived ADSC-SVF in Tendinopathy and Tendon Injury

In the literature, eight studies overall were present. The studies are listed in Table 2. Three studies were performed in rotator cuff disease. Jo et al. [49] compared the clinical, arthroscopical and radiological outcomes in a crescent dose of ADSC intralesional administration in patients with a partial-thickness rotator cuff tear tendon. A significant alleviated shoulder pain was shown, and a higher overall function of the shoulder was regained. Arthroscopic and MRI examination showed that the volume of the bursal-side defect significantly decreased from 90 to 83%. Kim et al. showed that during arthroscopic rotator cuff tear repairs, an injection of ADSC loaded in fibrin glue could significantly improve structural outcomes in terms of the retear rate at MRI analysis. However, there were no clinical differences in the 28-month period of follow-up [50]. Hurd et al. [51] compared patients with partial thickness rotator cuff tears treated with peritendinous corticosteroid injection and ADSC injection. At the final follow-up, a significantly higher American Shoulder and Elbow Surgeons Shoulder Score (ASES) total score was found in the ADSC group.

Three studies were performed in lateral chronic elbow tendinopathy (LET). Freitag et al. reported a case report of common extensor tendon chronic tendinopathy in a professional master golfer, which was treated with a peritendinous injection of ADSC and PRP. An immediate ameliorant of the clinical score was found and maintained during follow-up. Moreover, US and MRI investigations showed a regained normal tendon-like structure [52]. Lee et al. reported the clinical effects of allogenic ADSC injected in lateral epicondylitis through VAS, Mayo Elbow Performance Index (MEPI) score and US evaluation. The authors showed there was significant improvement in pain, functionality and anatomical defects [53]. Khouri et al. [54] treated chronic recalcitrant LET with a single injection of ADSC. VAS, qDASH evaluation and an MRI performed 6 months after treatment revealed a significant decrease in VAS and qDASh values. Moreover, the MRI score for tendinopathy improved significantly. Finally, Khouri et al. reported the effects of ADSC injection in patellar tendinopathy after a single administration of ADSC. A significant improvement in VAS, the Victorian Institute Sport Assessment-P (Visa-P) score and MRI tendon features was demonstrated [55].

**Table 2 medicina-59-00273-t002:** Enzymatically derived ADSC-SVF human tendinopathy and tendon injury.

Authors	Injury	ADSC-SVF Provenience	ADSC-SVF Delivery	Investigation	Outcomes
Jo et al. [49]	Partial thickness Rotator cuff tear	Autologous from abdominal area	Intralesional-injection	Clinical (SPADI score, CONSTANT score, VAS) Radiological (MRI), Arthroscopic	Significantly alleviated shoulder pain with more than 70% reduction from the baseline in the high-dose group. At MRI the bursal-side defect significantly decreased by 90% in the high-dose group . Arthroscopic examination volume of the articular- and bursal- side defects decreased by 83% and 90%.
Kim et al. [50]	Partial thickness Rotator cuff tear	Autologous, derived from buttocks	ADSC loaded in fibrin glue was injected compared to standard suture	Clinical (VAS, ROM, CONSTANT score, UCLA score) Radiological (MRI)	No clinical differences. At MRI, improved structural outcomes, low retear rate
Hurd et al. [51]	Partial thickness Rotator cuff tear	Autologous, derived from either the periumbilical abdominal area, bilateral flanks, or medial thigh fat	Intralesional-injection	Clinical (ASES score, SF-36)	Significantly higher ASES score
Freitag et al. [52]	Common extensor tendon chronic tendinopathy	Autologous abdominal fat	Intralesional injection enriched with PRP	Clinical (NPRT, PRTE) Radiological (US, MRI)	Significant improvements of clinical score and tendon structure
Khouri et al. [53]	Chronic lateral elbow tendinopathy	Autologous, derived from periumbilical zone	Percutaneous injection to the affected elbow	Clinical (VAS, qDASH) Radiological (MRI)	Improved VAS scores for maximum pain score, QuickDASH -Compulsory score, QuickDASH-Sport score, MRI score for tendinopathy.
Lee et al. [54]	Lateral epicondylosis	Allogenic	Intratendinous injection with fibrin glue	Clinical (VAS, MEPI score) Radiological (US)	Safe and improved elbow pain VAS, performance MEPI score, and structural defects
Khouri et al. [55]	Patellar tendinopathy	Autologous, derived from periumbilical zone	Percutaneous injection to the affected tendon	Clinical (VAS, Visa-P) Radiological (MRI)	VAS and Visa-P score improved significantly. MRI revealed significantive improvement for tendon thickness, and tear length, width and thickness compared to baseline.

ADSC (Adipose-Derived Stem Cell); SVF (Stromal Vascular Fraction); VAS (Visual Analogue Scale); MRI (Magnetic Resonance Imaging); Shoulder Pain and Disability Index (SPADI); University of California at Los Angeles—Activity (UCLA); American Shoulder and Elbow Surgeons (ASES); 36-Item Short Form Health Survey (SF-36); Numeric Pain Rating Scale (NPRT); Patient-Rated Tennis Elbow Evaluation (PRTE); Mayo elbow performance index (MEPI); Victorian Institute Sport Assessment-Patellar score (VISA-p).

#### 3.2.2. Mechanically Derived ADSC-SVF in Tendinopathy and Tendon Injury

Three studies are present and listed in Table 3.

Striano et al. [56] injected micro-fragmented adipose tissue harvested from the abdomen and processed with Lipogems^®^ (Lipogems USA, Atlanta, GA, USA) patients with varying cuff pathology. At the 1-year follow-up, a significant improvement in National Pain Scale (NPS) and American Shoulder and Elbow Surgeons Score (ASES) was reported.

Two studies are about Achilles tendinopathy. Albano et al. [57] investigated the correlation between US and MRI changes and clinical outcomes in Achilles tendinopathy treated with PRP or ADSC/SVF single injection processed with Hy-tissue SVF^®^ (Fastkit) (Fidia Farmaceutici, Abano Terme, PD, Italy). Six months after treatment, a significant decrease in VAS and an increased tendon thickness was found in both groups compared to the baseline. Usuelli et al. [58] compared the efficacy of PRP with the stromal vascular fraction of ADSCs (SVF) in 44 patients with Achilles tendinopathy in a randomized, controlled trial (RCT). All adipose tissue was processed with the Hy-tissue SVF system. After treatment, a significant, immediate benefit for the SVF group compared to the PRP group when analyzing the VISA-A, VAS pain and American Orthopaedic Foot and Ankle Society’s (AOFAS) score at 15 and 30 days was found. However, this effect did not persist past this time point.

All the analyzed studies performed in humans revealed, even in enzymatic or in mechanic ADSC processing, a consistent pain reduction associated to better functional outcomes. Moreover, US and MRI evaluation revealed an improvement in tendon structure and inflammatory signs.

### 3.3. Biomaterials Study

Several authors described the worthwhile association between ADSC/SVF and biomaterials in vitro [59,60,61,62,63]. However, for this specific review, four articles have been selected for the combined application of biomaterials and ADSC-SVF in tendinopathy and tendon injury in vivo (Table 4).

Lipner et al. [64] in 2015 examined the effect of an aligned nanofibrous poly lactic co-glycolic acid scaffold (N- PLGA) seeded with allogenic ADSCs and implanted at the repair site of rotator cuffs in rats. The healing response was examined in four groups (suture only, acellular scaffold, cellular ADSC scaffold and cellular ADSC + BMP2 scaffold) using histologic, bone morphology and biomechanics outcomes. The acellular scaffold group showed a delayed healing response compared to other groups, while cellular + BMP2 scaffold and scaffold in general showed decreased mechanical properties. Bone mineral density (BMD) was not significantly different among the groups. Chiou et al. evaluated decellularized tendon hydrogel (tHG) effects enriched with PRP and ADSC in Achilles tendon defects in mice. They demonstrated that the combination of PRP and ADSC show the organization of higher fibres, while no differences were found in biomechanical testing [65]. Deng et al. [66] used polyglycolic acid (PGA) and polylactic acid (PLA) scaffold enriched with ADSC in Achilles tendon defects in rabbits. At the final follow-up, a higher mechanical strength was demonstrated in the ADSC group, as well as a better histological score. Franklin et al. in 2020 demonstrated the homing of systemic ADSC in proper human decellularized tendon derived from hydrogel scaffold, in rats’ Achilles tendon defects. Flow cytometric analysis revealed a higher homing of ADSC in the injury site in the treated group [67].

A steady improvement in histological analysis was found from these studies. However, there is no consensus in the biomechanical result.

## 4. Discussion

Tendon healing required the participation of numerous cytokines, such as interleukin (IL)-6, IL-1β), growth factors (e.g., basic fibroblast growth factor [bFGF], TGF-β1, IGF-1, platelet-derived growth factor [PDGF], and vascular endothelial growth factor [VEGF]), and BMP-12,13,14, which are released in a specific temporally and spatially controlled manner [68,69,70]. However, this extremely delicate mechanism can be easily overturned, impairing the healing process.

Currently, the most common treatment for tendon injury is surgical treatment and repair. Unfortunately, despite the available treatment, issues with incorrect and slow tissue healing persist, resulting in reduced mechanical strength and suboptimal functional outcomes [65].

Regenerative medicine revolutionizes the treatment of musculoskeletal pathologies. MSC application gained increasing interest in the scientific community for the peculiar regenerative abilities in different tissues. Several studies demonstrated that MSC and, therefore, ADSC in a tendon injury are able to modulate the gene expression of the surrounding cells with a paracrine action through miRNA secretion and provide relevant exosome involvement in the benefits of MSC-based therapy [71,72]. Exosomes are extracellular vesicles (EV) with a diameter measurement ranging from 30 to 150 nm. In the course of multi-vesicular body development, an inward budding of endosomal membranes is observed, which contributes to the inter-cell communication. The multi-vesicular body endosomes fuse with the cell membrane, leading to the secretion of exosomes [73,74]. Most of the MSC paracrine factors are crucial to tissue regeneration and lined to the discharge of EVs. MSC exosomes originate in adipose tissue, the bone marrow and other tissues and bear a rich and complex load of nucleic acid (mRNA and miRNA), proteins and lipids [29]. 

They possess anti-inflammatory elements, reaching the recipient cells and decreasing inflammation. In view of these considerations, MSC-derived exosomes may be used in various inflammatory diseases such as tendon injury and tendinopathy.

Recent studies demonstrated that ADSCs are able to differentiate in tenocyte in vitro, and their application seems to aid in preserving the architecture of the native tendon tissue over time in culture through a faster and specific ECM remodeling [1,6].

Moreover, ADSC and SVF enhance human tendon cell viability, proliferation and growth factor production [75], with a significant upregulation of matrix degradation and downregulation of pro-inflammation-related factors [76]. Moreover, ADSCs have been shown to be immune privileged with a low risk of rejection and be more genetically stable in long-term culture, with a greater proliferative rate than BM-MSCs and minimal morbidity of the donor site [77,78,79,80]. Currently, many different formulations and protocols to prepare ADSCs/SVF are available, with little empirical evidence supporting or rejecting their use. Recent papers demonstrate that enzymatic or mechanical procedures are equivalent to obtain ADSC and SVF [26,27].

The advantage of the mechanical procedure is legislative, especially for European countries. In fact, according to the current European legislation, ADSCs must be mechanically isolated for human use. In fact, methods that provide for enzymatic digestion involve the destruction of the contact between the cells and represent a manipulation [81]. Even though mechanical procedures are faster and cheaper compared to enzymatical procedures, the disadvantages consist of the need to use a large quantity of lipoaspirate and the low viability of the resulting isolated ADSCs, thus making the non-enzymatic methods less efficient than the enzymatic methods [82,83].

Several animal models have been used to demonstrate the efficacy of ADSC/SVF in tendon injury (rats, rabbits, dogs, horses, sheep) [34,35,36,37,38,39,40,41,42,43,44,45,46,47,48]. The enzymatic procedure was the most commonly used. Achilles tendon injury was the most frequent recreated defect [34,35,40,41,45,46,47], followed by rotator cuff injury [36,37,43] and SDTF injury [38,39,44]. All the papers examined showed higher histological signs of structural tendon organization (number and orientation of collagen fibres), increased tenocyte viability, a higher expression in regenerative factors expression (COL-I, LOX, Dcn, TGFb1,) higher vascularization (VEGF, FVIII), and a reduced expression of scarring and proinflammatory factors. No consensus in biomechanical outcomes, such as tensile strength, was reported by the authors.

Human studies performed in extra-European countries revealed that the enzymatic procedure was the most commonly used. Rotator cuff disease was the most commonly studied [49,50,51], followed by chronic lateral tendinopathy [52,53,54]. All the studies examined showed immediate clinical benefits after ADSC/SVF application with increased functional outcomes and reduced pain. Imaging (US-MRI) showed the restoration of normal tendon conformation, with less swelling, oedema and thickness.

In European countries, two studies about Achilles tendinopathy [57,58] and rotator cuff tendinopathy [56] have been reported. As with the aforementioned studies, positive immediate clinical and functional outcomes have been reported.

No side effects or signs of rejection have been reported in all of the studies examined. Concerning biomaterials and innovative scaffolds are examined in vivo as stable supports for ADSC/SVF, especially in high tendon tissue defects [64,65,66,67,68]. Some of the newest scaffolds include N-PLGA, PGA/PLA and tHG decellularized tendon scaffold. They act as a 3D scaffold seeded with cells that can be directed to form tendon tissue. An enzymatic procedure was used in all the studies to obtain ADSC/SVF. While better histological immunohistochemical findings have been reported unitarily, no consensus about biomechanical outcomes were also present in this case.

Several critical issues must be highlighted.

First of all, legislative differences between European and non-European countries concerning “minimal tissue manipulation” prohibit enzymatic processing in Europe.

Moreover, there is no uniformity in ADSC/SVF preparation, neither enzymatical nor mechanical. Therefore, the literature is lacking about empirical evidence in supporting a specific technique. There is no evenness in ADSC/SVF harvesting, injectate content and frequency (i.e., cell count, use of mechanical or enzymatic isolation, use of additional injectates), use of US guidance during the procedures, and outcome and rehabilitation protocols to allow for the determination of optimum shared protocols.

Nevertheless, few randomized control studies are available at the moment, either in animals or in humans.

The main goal in this promising field for the future is a standardized characterization, preparation and processing of ADSC/SVF. A standardized and commonly shared therapeutic approach for animal and human tendon lesions in order to reduce the risks of clinical biases are required with large perspective double blind randomized control studies.

## 5. Conclusions

Taking together the findings of the present systematic review demonstrated that ADSC/SVF administration yielded positive effects in the treatment of tendon injury and tendinopathy, both in animal and human studies.

Current literature demonstrates that the clinical application of SVF and ADSCs is safe and feasible, even in humans.

Further efforts are needed to confirm the role of ADSC/SVF for the treatment of tendon injury and tendinopathy.

## Figures and Tables

**Figure 1 medicina-59-00273-f001:**
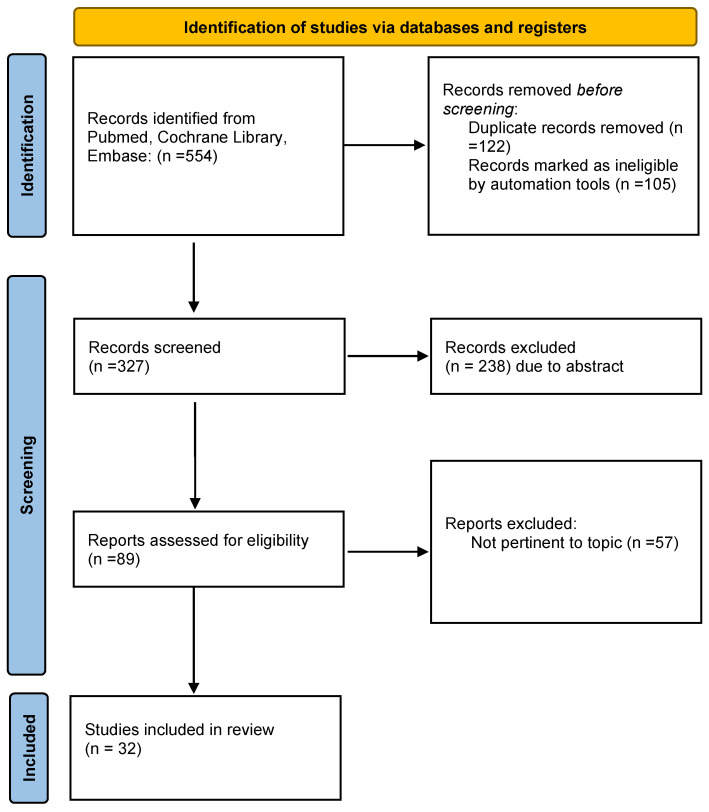
PRISMA flow diagram highlighting paper selection and search strategy.

**Table 3 medicina-59-00273-t003:** Mechanically derived ADSC-SVF human tendinopathy and tendon injury.

Authors	Injury	ADSC-SVF Provenience	ADSC-SVF Delivery	Device Used	Investigation	Outcomes
Striano et al. [56]	Partial thickness Rotator cuff tear	Autologous from thigh area	Intralesional-injection	Lipogems	Clinical (ASES score, NPRT)	Significant improvements in pain, function and quality of life
Albano et al. [57]	Chronic achilles’ tendinopathy	Autologous, derived from abdomen	Intralesional-injection	Hy-tissue SVF (Fastkit)	Clinical (VAS) Radiological (MRI, US)	Significant decrease VAS and an increased tendon thickness
Usuelli et al. [58]	Chronic achilles’ tendinopathy	Autologous, derived from abdomen	Intralesional-injection	Hy-tissue SVF (Fastkit)	Clinical (VAS, Visa-A, AOFAS)	Immediate benefits for VAS, Visa-A, AOFAS score

ADSC (Adipose-Derived Stem Cell); SVF (Stromal Vascular Fraction); VAS (Visual Analogue Scale); MRI (Magnetic Resonance Imaging); American Shoulder and Elbow Surgeon [ASES] Score; Numeric Pain Rating Scale (NPRT), Victorian Institute of Sport Assessment-Achilles score (VISA-A); American Orthopaedic Foot and Ankle score (AOFAS).

**Table 4 medicina-59-00273-t004:** Biomaterials and enzymatical/mechanical derived ADSC-SVF in vivo tendinopathy and tendon injury.

Authors	Injury (Animal)	Biomaterial	Treatment	ADSC-SVF Provenience	Investigation	Outcomes
Lipner et al. [64]	Rotator Cuff tears (Sprague Dawley rats)	Nanofibrous poly lactic co-glycolic acid scaffold	Group 1: tendon suture Group 2: tendon suture + acellular scaffold Group 3: tendon suture + ADSC scaffold Group 4: tendon suture + ADSC + BMP2 scaffold	Allogenic subcutaneous adipose tissue mice	Histological, bone morphology, biomechanical outcomes	The acellular scaffold showed a delayed healing response. Cellular + BMP2 scaffold showed decreased mechanical properties. No difference in bone morphology.
Chiou et al. [65]	Achilles tendon midsubstance defect (Wistar rats)	Tendon hydrogel scaffold	Group 1: saline control Group 2: tendon hydrogel Group 3: tendon hydrogel scaffold + PRP Group 4: tendon hydrogel scaffold + ADSC + PRP	Allogenic subcutaneous adipose tissue mice	Histological, Biomechanical analysis	Group 4 higher fibres organization. No biomechanical differences
Deng et al. [66]	Achilles’tendon defect (new zeland rabbits)	polyglycolic acid (PGA)+ PLA (polylactic acid)	Group 1: scaffold Group 2: scaffold + ADSC	Autologous, derived from nuchal side	Gross view, Histological Biomechanical analysis	Significant difference in mechanical strength in group 2, improved histological score.
Franklin et al. [67]	Achilles’tendon defect and rotator cuff chronic tendinopahty (Sprague Dawley rat)	Human decellularized flexor tendon hydrogel (tHG)	Group 1: saline Group 2: tHG Group 3: tHG + ADSC Group 4: tHg + FB Group5	In commerce	Flow citometry	Enhance homing of adsc cells administred ev.

## Data Availability

Not applicable.

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
