# Peer review of "Efficacy of Adipose-Derived Mesenchymal Stem Cells and Stromal Vascular Fraction Alone and Combined to Biomaterials in Tendinopathy or Tendon Injury: Systematic Review of Current Concepts"

_medicina, 2023, doi:10.3390/medicina59020273_

Round 1

Reviewer 1 Report

In this manuscript, the authors tried to provide a review of efficacy of adipose derived mesenchimal stem cells and stromal vascular fraction in tendinopathy or tendon injury.  The results show that ADSC/SVF therapy can represent a promising alternative in tendon regenerative medicine for the enhancement of tendon healing. In general, it's a well-written manuscript and provides a comprehensive update of the studies. However, there're still several issues which should be addressed.

1. In the section of "Results": The authors should add their own comments after the introduction of the literatures. 

2. In the section of "Discussion": The outcome and mechanism of ADSCs should be discussed in detail (Differentiation into other cells? Or just paracrine?)

3. In the setion of "Discussion": The author should discuss the main problems and limitations that currently exist.

4. In the section of "Discussion": The author should enumerate future research directions in detail.

Author Response

Dear Reviewer, thanks for your suggestions.

  1. In results section the comments of the authors have been added as requested (Line 221-227; line 282-285; line 316-317).
  2. The mechanism of ADSC has been added as required (line 343-361)
  3. The main current problem has ben added in discussion section as suggested (line 406-416)
  4. Future perspective has been added in discussion section (line 417-420 .)

Reviewer 2 Report

The systematic review from Letizia Senesi and others covers a topic of great interest and helps clarifying the role of ADSC in tendon disease. The argument is still full of variables in terms of methods of experimental design on the animal models, cells and tissue preparation, mode of delivery etc. Nonetheless, the authors are able to address the issue and obtain a useful take-home message for the clinicians. 

The paper is worth publishing. 

Author Response

Dear Reviewer, thanks a lot for your kind observation.

Reviewer 3 Report

This manuscript aims to discuss the treatment effect of ADSC and SVF on tendon regeneration. Overall the article is well in writing. The reviewer has the following comments that may be helpful to improve this article.

1. Even though the cellular study was not included in this study, the author can provide a short paragraph in the Discussion to mention the current study status in this field.

2. Even though this article focused on ADSC and the cells derived from the related tissue, a short paragraph can be provided in the Discussion to mention the current studies using other stem cells.

3. On line 117, "AMSC" should be "ADSC".   

the systematic review aims at providing a comprehensive update of the literature on clinical ap-23 plication of ADSC enzymatically or mechanically processed to obtain SVF, alone and in association 24 with biomaterials in the local treatment of tendinopathy and tendon injury in vivo, in animal models 25 and humans.

Author Response

Dear Reviewer, thak you very much for your suggestions.

  1. 2. The short paragraphs concerning the general status of cellular studies  and about MSC application has been added as requested (line 343-358). I hope it goes well.                                                                                                           3. The typo has been corrected as you suggest.

Round 2

Reviewer 1 Report

The revised version is better than the original one and should be accepted for publication.